# Designation, Incentivisation and Farmer Participation—Exploring Options for Sustainable Rural Landscapes

**John McDonagh**

Department of Geography, School of Geography, Archaeology & Irish Studies,
National University of Ireland (NUIG), H91 TK33 Galway, Ireland; john.mcdonagh@nuigalway.ie

**Abstract:** This paper explores how policies, management practices and farmer participation can help shape resilient and sustainable rural environments. The key elements of the paper draw together policy initiatives on land designation for conservation, action-based and results-based agri-environment programmes, and locally led and inclusive partnership models. In doing this, the paper explores ways to address environmental decline, while also allowing farmers to farm and for management practices to be developed that farmers can readily endorse. The paper draws from empirical evidence gathered from two case studies in West Ireland. These studies include in-depth interviews, consultations with key stakeholders and an exploration of policy and other documentation associated with the management of rural landscapes. What emerges from the discussion and the field evidence is that, while there can be discontent, even disillusionment with some practices, there are models of great promise evolving. In particular, the research identifies the importance of enabling a space in which a farmer's knowledge and expertise have due prominence and where they are afforded recognised input in the schemes being developed and promoted. The conclusion of the paper suggests that, while impacts vary, it is clear that combining forces from top-down and bottom-up, allied to locally led decision-making input, provides the Special Areas of Conservation combination whereby landscapes can be both farmed and protected.

**Keywords:** Special Areas of Conservation; designation; results-based payments; farmer participation

## 1. Introduction

The rural is at a time when opportunities are plentiful and numerous threats and challenges are ever-present. The rural is talked of as a panacea for addressing major challenges of climate change and food and energy security, while also being looked on as the main contributor to climate change, threats to biodiversity and broader ecosystem destruction. While both carry merit, the rural, for all its missteps and mistakes, is central to any possibility of a sustainable future. The key lies in the ambition and will to make decisions that facilitate a path toward a resilient future. In this paper, experiences from Ireland in terms of how we might envision sustainable rural landscapes going forward are explored with the aim of outlining ways in which policies dealing with landscape management practices, particularly as they relate to farming, can be more carefully constructed and result in successful rollout and acceptance by those charged with operationalizing them on the ground. What is also evident is that the underpinning concept that is referred to as a sustainable landscape needs to be appreciated in terms of its dynamism and continual alteration, as well as how its meaning, significance and management changes over time [1]. This paper then engages with farmers on the ground to identify how it is possible to develop 'intelligent decisions' [2] about the future of rural landscapes and, while not claiming to be the only pathway, presents an approach that demands serious consideration in terms of sustainable landscapes and what that might mean going forward.

The interesting thing about placing this research in the Irish landscape is its mosaic of topographies—rich agricultural lands, mountainous disadvantaged areas and internationally recognised Special Areas of Conservation—a past of over-exploitation, degradation and pollution and, in more recent times, a re-emergence of a new appreciation for protection and management of rural landscapes and their innate fragility. This paper, contextualized by Selman's dimensions of sustainable landscapes and particularly those focused on environment, economic, social and governance aspects [2] explores three aspects of recent Irish landscape management practices, namely that of Designation, whereby lands are declared as Special Areas of Conservation (SACs) (see Table 1 for a list of acronyms used in the text) and all that that entails; Appreciation—where some recent initiatives have sought to engage more closely with local farmers and allow their knowledge and input to influence how rural landscapes are managed; and lastly, Incentivisation, where results-based payments play a central part in directing farming toward more environmentally friendly practices and management systems. What emerges is that, while impacts vary, it is in combination, under locally led decision-making models, that the best solutions are found.

**Table 1.** List of acronyms.

| AES | Agri-Environmental Schemes |
| --- | --- |
| EIP | European Innovation Partnership |
| EIP-AGRI | European Innovation Partnership for Agricultural productivity and Sustainability |
| GLAS | Green Low-Carbon Agri-Environment Scheme |
| IFDL | Irish Farmers with Designated Land |
| LLAES | Locally Led Agricultural Environmental Scheme |
| NPWS | National Park and Wildlife Services |
| SAC | Special Area of Conservation |

## 2. Contextualising Rural Landscape Management

The rural occupies a central place in the European narrative. At its most basic, the agenda of providing safe and quality food and the expectation of a reasonable standard of living is constant, couched as it is in broader acceptance of sustainability principles and the need to be mindful of the needs of future generations. Indeed, McDonagh [3] argues that 'Europe and beyond are struggling with the twin challenges of producing safer quality foods, while preserving, if not enhancing, the natural environment in which it is produced' (p. 6). In terms of the crossroads that we find ourselves at currently, the decision of which direction to take is fraught with complexity. Somewhat similar to the ease with which support can be proclaimed for the principles of sustainability, it is less clearcut when it comes to enacting measures on the ground. This is particularly so when meeting the needs of the present can often outweigh the necessity to make unpalatable decisions that are for future well-being. Nowhere is this epitomised more than in our rural landscapes. Rural areas provide the ingredients for human survival, yet they are also those most threatened by human activity. Indeed, the management of rural landscapes has never been straightforward and can best be thought of as vacillating from extremes of great concern to that of reckless overuse [4].

Sustainability, as a consequence, has become the central tenet of contemporary policies relating to the use and management of rural resources. Pressure in recent years has ratchetted up considerably in terms of demands for a more sustainable management trajectory for rural areas and its role in issues such as climate change, in particular. Agriculture and agricultural practices have, perhaps, the most profound impact on the myriad of habitats and species that are found across our rural areas [5–8]. In fact, it could be argued that there is a strong interdependency between the protection and conservation of habitats and species and the type and intensity of the farming systems engaged in it. What is equally valid is that, in recent decades, the incongruity that is farm intensification and land abandonment has dramatically altered our biodiversity and broader ecosystems [8]. Indeed,

it took the economically unsustainable and politically untenable situation of continuing to subsidise farmers' augmentation of existing overproduction to bring about adjustments to EU agricultural policy [6]. This brought forth incentives to promote not only pluriactivity, particularly in the form of on-farm diversification, but also efforts to encourage more environmentally friendly farming practices. This, for the first time, led to measures for financial benefits being provided to farmers who were willing to adapt their farming to nature conservation requirements. This change in mindset was a major step forward, although it can be somewhat tempered in that farmers that availed of these measures were often those already conducting environmentally friendly farming practices in the first place.

In all, while initially in small steps, the greening of the agricultural agenda began with the objectives of making farming both multi-functional and profitable. A key component, and what has often been described as one of the most impressive achievements in the environmental field, was the emergence of agri-environmental policies [7–9]. What was apparent here was how the decline in extensive or traditional farming practices had placed a downward pressure on habitats and rural biodiversity. The drive to produce more had seen agricultural outputs reach unprecedented levels with consequent surpluses being put into intervention or destroyed [6]. While this continued unchecked for a number of years, in recent times, and particularly in the last decade, we have seen efforts to realign where agriculture is going, prompted by a growing concern for habitats and species decline and climate change. Indeed, the European agenda has, if not quite pivoted, reaffirmed its desire to one of 'attempt(ing) to manoeuvre agriculture to a more acceptable position with farmers being cast as custodians of the countryside' [10] (pp. 713–714). There have been a number of ways in which this realignment has played out on the ground. Undoubtedly, agri-environmental policy has been fundamental in creating numerous positive environmental responses to agriculture, including growing knowledge of nutrient and habitat management and educating toward greater environmental awareness. This process was not without its critics, however. The European Court of Auditors, for one, called for more targeting of agri-environment payments, an observation that was heeded in the development of Locally Led Agricultural Environmental Schemes (LLAESs), which bring a much-needed impetus to landscape management practices and a contribution that is examined later in this paper.

## 3. Ireland's Approach to Landscape Management

Ireland is a prime example of the challenges facing farmers and their sustainable management of the resources on which their livelihoods depend. Pressures associated with agriculture have had major impacts on land-based habitats and species, with over 70% of the number of habitats of EU interest reported to be negatively impacted by agriculture [5]. Ecologically unsuitable grazing regimes and abandonment are the main Irish pressures reported, with the Irish National Biodiversity Plan 2017–2021 [5] declaring that the breeding populations of bird species that are associated with farmland, such as the Curlew, Lapwing and Yellowhammer, have declined substantially over recent decades, some to the brink of extinction [10]. From such a starting point, the challenge in terms of conservation being integrated into sustainable agricultural practices is significant.

Intensification, a key aspect of the modernisation of agriculture, had the unfortunate side effect of increasing pressure on the environment. Consequently, the reforms of the EU's Common Agricultural Policy (CAP) since 1992 have aimed to progressively reduce this pressure with several instruments and tools developed to mitigate the environmental impact of agriculture. Agricultural Environment Schemes (AESs) have been one of the foremost of these tools. Developing alongside the newfound interest in AES, a growing number of protected areas also emerged globally. As well as their development, what is also interesting is the way in which many of these areas are governed and managed, particularly as they are integral in bringing together environmental concerns and the practice of farming.

### 3.1. Special Areas of Conservation (SAC)

There are over 13,500 sq. km of SAC designations in Ireland covering all types of landscapes. The designation of SACs stems from the EU Habitats Directive, and they are part of the NATURA 2000 network of European protected sites. There are over 400 SAC designated sites in Ireland covering 13% of the land area [10]. The challenge for landowners, even if they want to protect the habitats on their land, is that they are not automatically entitled to be compensated for SAC being part of their holdings. They are, however, restricted in the type of activities they can engage in and are fined if they breach SAC restrictions. While a vast majority of SACs in Ireland are state owned (by the National Parks and Wildlife Service (NPWS), Coillte (Forestry Agency) and the ESB (Electricity Supply Board), for example), there are many farmers who are affected by this designation process. In one effort to address this, a National Farm Scheme was introduced in 2004 that provided compensation to SAC designated farms, but its decline in 2012 led to farmers being left with the 'burden' of SAC designation and being 'unable to farm as they wished and . . . unable to maximise their lands to their full potential' [11,12]. What is significant in this process, and a situation that undermines the role of farmers somewhat in terms of their role in managing the landscape, is how SAC designation is very much a top-down political and scientific decision-making process. Landowners are notified of the proposed designation, are sent an information pack explaining the scientific reasons for the designation and are given details of how they can go about appealing the process. What is readily apparent is the absence of any attempt at consultation between landowner and policymaker. This top-down approach excludes input from the farmer, leaving a situation of disconnect and disempowerment rather than trust and mutual respect. Discussions with farm owners reinforce the view of limited participation, which, if it did occur, came 'after scientific evaluations and decisions have been made (and carries) a strong top-down "conservation" imprint with less regard for its social acceptance and feasibility at a local level, although land designated is to be managed by farmers' [13] (p. 29). Indeed, the designations of SAC sites has often met with opposition from groups such as the Irish Farmers Association (the IFA is the largest farmer organisation in Ireland), with even a dedicated group called the Irish Farmers with Designated Land (IFDL) being set up. The aim of this group was one of uniting farmers and landowners in regaining the value of designated land and ensuring farmers could generate a reasonable income from designated lands [14].

### 3.2. Locally Led Agricultural Environment Scheme (LLAES)

One programme which has the potential to address the shortcomings of Designations and the broader remits of national Agricultural Environmental Schemes is that of the Locally Led Agricultural Environmental Scheme (LLAES). Prompted by EU policy, such schemes are intended to encourage locally driven solutions to address local issues. Theoretically, these schemes present a real opportunity to reach the spaces where knowledge is shaped and transferred. Ireland's Locally Led Agricultural Environment Schemes (LLAESs) are specifically targeted to meet the requirements of EU Birds, Habitats and Water Framework Directives and aim to address particular environmental and biodiversity challenges not addressed at national level by the Green Low-Carbon Agri-environment Scheme (GLAS) (this scheme was introduced under the Irish Rural Development programme 2014–2020 and provides payments to farmers to help tackle climate change, preserve biodiversity, protect habitats and promote environmentally friendly farming) [15]. The centrally identified priorities include the continuation of the BurrenLIFE programme, priority pearl mussel catchments and hen harrier areas. LLAESs encourage locally driven solutions and require submission of proposals by local groups accompanied by detailed estimates of costs. A great example is evident in the Burren Programme. This innovative programme takes a farmer led approach, where the farmer nominates and co-funds conservation actions on their farm, giving the farmer a type of freedom to farm. What makes the Burren different is that it combines these actions with a results-based payment. To ensure that the desired

results are achieved, payments are made to farmers based on the environmental condition of their farm. The better a farmer's field 'scores' in terms of environmental outcome, the more payment they receive. The way in which these scores are reached and implemented is that "eligible field are assessed annually by the farm advisor using a user-friendly "habitat health" checklist. Farmers are made aware of their scores (and) all scores are reviewed for accuracy and consistency by the Burren team and many also are checked by Dept. of Agriculture inspectors. The field score, which ranges from 0 to 10, is calculated using nine distinct, weighted criteria which, taken together, give a very accurate picture of the "health" of the grazed habitats in that management unit. These criteria are: Grazing level; Amount of litter (dead vegetation); Extent of feed site damage; Extent of damage at natural water sources; Level of bare soil and erosion; Level of encroaching scrub; Amount of bracken and purple moor grass; Extent of weeds and agriculturally-favoured species; and Ecological integrity. Once the field score is calculated, it is multiplied by the available payment rate per hectare and by the size (ha) of the field, to calculate the "output payment" for that field. Under the Burren Programme, all fields with a score of 6 or more receive payment but higher scores receive higher payments—increased payment rates are available for fields scoring 9s and 10s. Fields with a score of 5, only receive payment in the first two years. Payments per ha can range from €8/ha to €180/ha depending on field score and farm size (payments are "banded" to reward smaller holdings). This gives farmers the incentive to manage their fields in ways that will improve their scores and their payment as well" [16].

A key component is the way in which there is a partnership approach involving all the key stakeholders, farmers, state agencies and government departments in tailoring solutions in practical and applied ways.

Indeed, it has been suggested that LLAESs could be rolled out across EU Member States and variously adapted and applied to areas of High Nature Value (HNV) throughout [17,18]. This type of approach is now beginning to emerge as a key instrument in the new CAP 2023–2027, with Ireland, for example, instigating a new Results-Based Environment Agri Pilot (REAP), which seems to be generating a lot of interest from farmers, with 10,000 initial applications received thus far [19].

### 3.3. EIP and EIPAgri

The emergence of European Innovation Partnerships (EIPs) has stemmed from the broader EU strategy of 'Europe 2020', with its desire to ensure that EU citizens have increased employment opportunities, a better quality of life and a competitive economy in which to live [20]. The basic tenet is that of bringing together public and private sectors across all scales to co-operate through research, innovation and practice in building a better and more sustainable and inclusive economy. EIP-Agri, launched in 2012, focused on bringing together stakeholders engaged in agriculture and forestry sectors, with innovation and fostering co-operation between researchers and practitioners being fundamental. In particular, EIP-Agri has at its core Operational Groups that explore new insights while also drawing on existing tacit knowledge in a bid to find solutions to specific issues (agricultural and forestry sectors) and to develop new opportunities (ibid). Essentially, what makes this initiative a step in a different direction is that it provides funding (€59m) to bring together as many stakeholders as possible, with the aim of dealing with a local issue locally [20]. When EIP-Agri was launched in Ireland in 2016, there was a large volume of interest and a large array of projects and proposals submitted. There are currently 23 projects operating on the ground, and their impact has been hailed a significant contribution in how we manage our rural landscapes. Indeed, EIP-AGRI projects have been described as being central to addressing 'challenges such as biodiversity, profitability and sustainability (while) harness(ing) the creativity and resourcefulness which is the hallmark of Ireland's rural sector' [21] (p. 3).

### 3.4. Results-Based Payments

There is a shift to move beyond compensating farmers for halting negative practices and instead incentivise positive management by paying for the delivery of clearly defined, measurable environmental outputs/results. Wynn-Jones [22] suggests that results-based agri-environment schemes should be seen as a 'new form of production' (p. 77) rather than an offshoot of agriculture. In essence, the results-based approach looks to instil behavioural change in what a farmer does through adapting to the specific demands of different areas and through famers being centrally involved in the design and development of how environmental objectives should be attained. This approach instils a greater sense of ownership among farmers and provides a strong platform on which environmentally positive farming practices can be built. The payment structure of agri-environment programmes can be divided into two main categories: outcome/results-based payments and prescription/action-based payments. Research and results from a number of pilot projects have shown that results-orientated agri-environment programmes offer a more effective means of delivering better environmental outcomes if they are well designed and are accompanied by robust environmental indicators to measure outcomes [23,24].

The positivity that is growing around a results-based payment system centres on how it allows farmers greater freedom to decide how to manage their land and use their skill and experience to improve environmental and agricultural performance. In addressing broader policy concerns, it also suggests a better-value-for-money model, in that, if deliverables are not met, then payment is not made. While one of the weaknesses of the action-based system is the lack of monitoring or ability to measure positive changes, results-based payments employ a field scoring system which generates data that help determine where positive environmental impacts are occurring [16]. Undoubtedly this approach requires a change in mindset of farmers toward how they farm. However, what is also apparent is that this new pathway very much calls into question the 'one-size-fits-all' approach favoured in EU policy to one that takes cognisance of location-specific needs and challenges. This approach not only demands flexibility and adaptability but draws on and incorporates farmer/local knowledge into its design and development. Consequently, what has emerged in the recent literature is that results-based approaches can not only add value to action-based ones, but 'can be adapted to complement action-based approaches and be geographically targeted to situations where they are best suited' [25] (p. 296).

## 4. Methodology

As well as drawing on broader research concerning aspects of land-use management, empirical evidence for this paper was also drawn from a series of interviews conducted in the Western Region of Ireland. The information gathered greatly enhanced an understanding of the broader discourse in the area of land management, use and protection. In all, 30 semi-structured interviews took place over a period of months during 2017/2018, and these were backed up by other documentation, including website materials, brochures and newspaper articles. The interviews were recorded and transcribed, and when used in the text, they are indicated by letter and number. Nvivo was used in relation to the data analysis, alongside a content analysis approach. The Western Region of Ireland affords insight into an unusual amalgam of land use, economic activity, conservation and sustainability (economic and social) challenges. The interviews with the landowners allowed us insight into and access to first-hand knowledge of, and dealings with, fragile landscapes and the farming practices therein. Two areas in the western part of Ireland, namely the Burren (comments from those interviewed are identified by the letter B and a number) and the region around the Ox Mountains (comments from those interviewed are identified by the letters Ox and a number), were the main study areas. The Burren is a particularly fragile landscape that is noted for its archaeology, flora and fauna interspersed by a considerable amount of farmed land, while the Ox Mountains has the typical challenges that come with a mixed topography, fragmented holdings and mixed-quality lands in terms of farming practices. The farmers talked to us, providing an array of viewpoints, with some being

more advanced in their own engagement with sustainable farming practices, that is, the various AES schemes available, than others.

## 5. Results and Analysis

In recent decades, various approaches have been made to halt the decline of biodiversity in agricultural areas; designating lands as protected areas and providing financial incentives to farmers to join agri-environmental schemes have been typical instruments.

### 5.1. Designation

In terms of land designation, the process has often met with conflict from landowners and disillusionment at the perceived top-down lack of consultation that this process has often entailed. The reaction to land designation in Ireland (and elsewhere) is often projected as a feeling of exclusion from the procedure, with farmers citing the absence of local input, knowledge, traditions and values in the drawing up of designations [26,27]. Consequently, many designations often result in, if not failure, only lukewarm acceptance. In the context of this study, the empirical results reinforced many of these aspects, with top-down decision-making, the complexity of the process and the type of prescriptive type of payment process all coming in for criticism.

SAC designation has tended to reflect top-down attempts at participatory or multi-stakeholder consultation and never fully embraced local knowledge and practices as valuable expertise in the sustainable management of these areas [28]. During the course of the fieldwork for this paper, there was undoubted support and understanding of the need to protect fragile landscapes and acknowledgement of historical, cultural and environmental significance in respective regions. However, the cursory inclusion of farmers as key stakeholders, as well as their peripheral placement in the design and development of SACs, engendered what could best be described as a sense of disillusionment. Indeed, in terms of land designations, there was a very emotional response in the interviews to this perceived lack of involvement, as well as a sense of frustration. Some farmers pointed toward the absence of local knowledge and the role it could play with one declaring that '*they're a bit useless for some farmers*' (Ox18), while another suggested that '*they don't make the most of what could be done on different areas of land. I don't have great things to say about them anyway*' (Ox8). There was also a sense of anxiety evident with regard to how restrictions were placed on private lands which conflicted with a farmers' desire to meet their own land-use objectives: '*There's another one near us, another SAC . . . I wouldn't like to see us being in one really because it would upset the land*' (Ox1). The additional costs associated with designation were also highlighted: '*It's a bit of a problem I'd say, especially young people going looking for planning permission now because it becomes an awful lot more expensive you have to get an archaeological report on the site, so it's a big problem*' (B1). Others referred to their own farm enterprise and the knock-on effect such designation on their lands would have in that there would be costs associated with not being able to develop their farms and '*make a decent living*' (B2), because, as one farmer also pointed out, '*you see . . . I can't really see how I can improve . . . if you compare . . . to somebody that isn't on the (SAC) site, they can do what they want so of course there's an economic impact*' (B1).

It was also felt by many of those interviewed in both study regions that many existing farming systems and practices were already compatible with environmental goals and that some of the new schemes were going against what they felt worked best: '*all of these rules now don't apply to the reality, they don't account for best practices*' (Ox3). One farmer pointed to how '*some of the schemes don't really understand what we're about*', while another referred to the knowledge that was handed on to him and which was now being ignored: '*one of my uncles taught me all I know about this mountain, what time of year to put sheep out. Now for GLAS we're forced to keep them out for seven months of the year . . . they don't want to know the fact that the grass only turns sweet at the end of May beginning June on a good year*' (Ox8). Many of the farmers interviewed pointed to difficulties with the top-down nature of the process and what they perceived as the '*endless form filling required*' (B7). In some of the conversations

during the field research, farmers suggested that '*these schemes need to be simple and flexible if they want farmers to buy into them*' (Ox4) and they '*need to suit the area*' (B6).

While the consensus among the farmers saw merit in what designations were trying to achieve, they also felt that, as the ones charged with delivering on these objectives, they had very little input into the design or decision-making around the process: '*we're the ones that should be front and centre ... we're doing all the work*' (Ox2). This was evident with farmers critical of the lack of dialogue between top-down and bottom-up in terms of existing policies and practices, with a farmer declaring that '*it's one way ... there's no question of tell us back what does and doesn't work for you. It is very much a one way street*' (Ox3). Another farmer reinforced this sentiment in his comment that there was a '*big disconnect between what the policy makers think they know and what was is really happening on the ground*' (B8). While the issues of payments and prescribed rules (and penalties) were a common thread in many of the discussions, it was also clear that the farmers wanted to be heard and appreciated for the knowledge and experience they themselves had accrued, as signified by the comments of one farmer who declared the following: "*if it was farmed with some of the practices that I know it would respond better and I'm under no doubt about that*" (Ox7).

The payment structure of the AES has often proven to be problematic in that it is based on prescription/action-based payments for the adoption of particular land uses or land management practices [29]. These scientifically defined criteria with prescribed sets of rules [17] do not, for the most part, account for local conditions or farmer knowledge. Consequently, frustration, if not anger, was evident during the field study in relation to the perceived contradictory nature of some policies and the subsequent knowledge that farmers receive. The often-punitive nature of the policies, that is, punishment for wrongdoings rather than incentivise for good, often left farmers frustrated. One farmer recounted how '*one fella ... took over a farm here and he was doing improvements on it and they stopped him, and ... they stopped his payments ... he was mak(ing) better walls ... but they wouldn't let him do that*' (B1). Another farmer referred to how there was a lack of communication between policymakers and what a farmer could engage with on the ground: '*you see ... penalties were imposed on the basis of insufficient farming activity, but the Department (of Agriculture) never set out a criteria for what they defined as insufficient farming activity, neither did they take into account the environmental condition of these lands or even carry out an appropriate assessment where lands had a designation on them*' (Ox7).

The example of the Burren programme is perhaps the catalyst for change that is necessary to ensure greater buy-in and support from farmers; many of the procedural obstacles were removed through innovative processes such as unique field scoring systems and simplified farm plan and paperwork, which all help '*minimise the bureaucratic burden*' [16]. This is essentially, the provision of a scheme that focuses on conservation results rather than strict management methods or prescriptions, as well as being one that is tailored for areas with differing farming systems, habitats and species.

### 5.2. Incentivisation and Participation

Studies show that polices tailored to the situation on the ground, with adjusting payments, as well as financial incentives, have positive influences on participation [30]. In the context of agri-environmental schemes, the approach has been very often couched in the belief that farmers' actions were primarily driven by an economic rationale, and, therefore, providing economic gains for farmers would lead to a change is practice [31]. The reality, however, proves somewhat different, with many farmers in the study areas being multi-generational and deeply imbedded in their rural communities and societies. While financial incentives may 'buy' some commitment, as one farmer commented, '*you don't farm in area like this for the money*' (B2).

The concept of Locally Led Agricultural Environmental Schemes (LLAESs) has, therefore, significant potential to bring innovative solutions to bear and to ensure sustainable land management. Likewise, EIPs promote local solutions to specific issues and involve the establishment of Operational Groups (OGs) to develop ideas or take existing ideas/research

and put them into practice by being hands on in terms of working toward the resolution of a practical problem. The results from the Burren Programme reinforce these observations with clear evidence of direct and indirect social, economic and environmental impacts being accrued. Farmers talked of the freedom they have been given whereby they use their own skills and experience to deliver on environmental needs, in addition to their farms benefitting from better management practices. This was also very much reflected in the comments of a farmer from the Ox Mountains region when he declared that '*farms should continue to be economically viable for the people farming on Ox mountain like it has been for generations*' (Ox7).

One of the basic principles of the Burren Programme was that the learning process was based on the participation of farmers in the process. They were the ones experimenting, evaluating and selecting practices and solutions that adapt best to their own farm's conditions. In this process, local tacit knowledge is not arbitrarily extracted; it is shared knowledge. Scientists, technicians and farmers work together and are coordinated through horizontal linkages. The best outcome and best possibility of getting buy-in from farmers seems to rest on their meaningful inclusion, and, as reflected in the comments of one farmer, they also '*want the area to be protected . . . I don't want an area to die . . . I want it to be a lived-in landscape*' (B1). This sentiment was also reflected in the comments of a farmer from the Ox Mountains region who suggested that '*you need to have all voices heard*' (Ox7). In place of generic criteria imposed through government policy and farmers in engaging in, what one described as '*a guessing game*' (Ox3), having farmers involved in the decision-making process from the initial design stage through to how each plan is derived allows the farmer to develop a sense of ownership and certainly has proven to deliver results [32].

The outcome-based approach sees the programme manager paying for results and, hence, not looking for breaches, ensuring a better working relationship and fewer non-compliance issues [17]. A devolving of power to farmers in terms of self-assessment on how they are preforming also has significant impacts in terms of the level of trust between top-down and bottom-up. In addition, a 'personalised' designing of individual farm plans, with advisor and farmer working hand in hand, sees the farmer's knowledge being greatly valued and given status through its incorporation into the design of their farm plan. All of these measures are extremely positive steps going forward in terms of addressing the issues of disconnect felt between farmers and policymakers and the sense of exclusion felt by farmers in terms of being able to contribute to policies that they are expected to deliver on. This type of integrated strategy was described by one farmer as being 'essential' and a necessary requirement to '*drive that forward . . . so you can create and implement structures that will foster local communities, and local communities here (are) centred around farming and that shouldn't have to change*'(Ox7).

## 6. Discussion and Concluding Remarks

During the course of this discussion, a number of aspects have come to the fore, and while the evidence presented here reflects the possibilities within an Irish landscape, the message of positioning a farmer's input and knowledge in a prominent position in the designing of agri-environmental programmes is applicable across Europe and beyond. While by no means extensive, the following have thus emerged as key ingredients in the pursuit of sustainability pathways for agriculture in the coming decades:

### 6.1. The Importance of Multi-Stakeholder Involvement and a Prominent Role for Farmers in the Decision-Making Process

There is a realisation that, while direct intervention can have a specific impact in the drive toward biodiversity conservation, the inclusion of farmers is an essential component. While the propensity has been for a top-down scientific driven model, multi-stakeholder involvement is paramount. The willing participation of farmers, who, for the most part, are the main landowners, and a sense of ownership of policy measures that impact their practices, is vital to the effective implementation of any landscape management approach.

This was very evident in the practices found in the Burren, where there was a strong rapport between stakeholders with the input of the farmer carrying weight and value. The conversations and data from the Ox Mountains area, however, suggested that there was still some way to go to achieve such inclusion and partnership. Indeed, it could be argued that the Burren is more the exception with the Ox Mountains region more reflective of the broader challenges still to be addressed. In particular, many of the reflections and discussions with the farmers suggested indications of disconnect between the various stakeholders and the vision that was trying to be moulded. In fact, the integration of local experiences, scientific knowledge and farmers' ideas into policymakers' demands was in its infancy in terms of the requisite trust and partnership required. In many comments, there seemed to be a sense of powerlessness among local landowners in terms of the decisions being made about how the landscape should be managed and a feeling that there was negligible recognition of their role, with continued debates on who, when, where and what knowledge are included [33]. Although national AES remains a science-first or ecology-first process [34], the LLAES demonstrates a potential to provide a platform on which scientists, policymakers and farmers can work toward a common goal in terms of better environmental practices and land stewardship.

### 6.2. The Combination of Action-Based, Results-Based and Locally Led Programmes

The demands for environmental protection and management have never been greater. The necessity to ensure this needs the appropriate architecture that will enable such a pathway to evolve. The combination of action-based and results-based approaches, allied to locally adapted practices, can drive this change. This combination will invariably enhance environmental practices. Perhaps the most striking aspect of this new pathway very much calls into question the 'one-size-fits-all' policy direction currently dominating, to one that takes cognisance of location-specific needs and challenges. While the specific attributes of any given place demand a certain type of approach, there are many commonalities that can be operationalised across all EU member states. The combination of action-based, results-based and locally led programmes provides flexibility and adaptability, while also drawing on and incorporating local farmer knowledge in design, development and implementation. The Burren Programme reflected this particularly well in that it adopted a hybrid outcome-based payment system, with two main measures that absorb roughly equal funding, one for actions (capital works) and the other for outputs/results [32]. The positive outcomes thus far point to a process that enables an alternative means of achieving environmental objectives.

### 6.3. Integrating Local and Scientific Knowledge in Pursuit of the Best Environmental Outcomes

If we accept that 'knowledge-sharing and community-learning processes . . . contribute to sustainability' [35] (p. 257), then the absence of such practices is surely undermining efforts at developing a sustainable future. McDonagh et al. [13] suggested that there is 'often (a) contradictory nature of top-down policies that frustrate those on the ground and in many instances create unnecessary tension and conflict' (p. 122). The challenge for policymakers is one of how to engage local people and how to extract their 'tacit and embedded knowledge' [36] and not in a 'box-ticking' way or one that does not 'recognise the value this can add to decision-making related to landscape and natural resource use' [13] (p. 126). Indeed, it is hard to envisage the landscapes of the two study regions explored in this paper being sufficiently maintained without 'the land management and livestock husbandry skills of farmers and the cultures of their communities' [37] (p. 90). Consequently, the important role played by the farmers' experience and the knowledge they possess cannot, and should not, be dismissed or underestimated. Indeed, a key aspect of the activities in the Burren region is very much about demonstrating to, informing and listening to farmers instead of imposing restrictions. The Burren Programme, for example, consciously incorporated these insights into its programme, with locals and part-time farmers given extensive training and then being hired as advisors to work for the Burren Programme.

This has the important outcome of developing a sense of ownership within the community and, more important, increasing the connection between landowner and policy objectives.

In a final comment, there is no doubt that the decline in global biodiversity is at a critical level, and the reduction in the biodiversity on our farms is a major contributor to this. What has been presented here is the significant role that carefully crafted and inclusive agri-environment schemes can deliver. An opportunity which can be a key instrument is addressing biodiversity decline and one that can fashion a pathway toward greater resilience and sustainability in our rural landscapes.

**Funding:** This research received no external funding.

**Institutional Review Board Statement:** Not applicable.

**Informed Consent Statement:** Not applicable.

**Data Availability Statement:** Not applicable.

**Conflicts of Interest:** The author declares no conflict of interest.

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
