# Peer review of "Designation, Incentivisation and Farmer Participation—Exploring Options for Sustainable Rural Landscapes"

_sustainability, doi:10.3390/su14095569_

Round 1

Reviewer 1 Report

A very interesting and important study. It demonstrates that the knowledge of farmers and their input in the initial stages of a designation are important. Aspects that could do with being addressed mainly concern the structure of the paper. 

Abstract reads more like an introduction than a summary of the paper. Please outline introduction, methodology, results, discussion and conclusion. This is important when systematic reviewers search for papers. A good abstract is useful. 

The introduction would benefit from better referencing. There are only three references in section 2 and none in section 1. 

There are quite a few acronyms. A table would help in the understanding of the different schemes and how they relate to farming in these areas, so acronym, long name and short description of perhaps one sentence. Also an indication of the origin of the scheme, local, national or EU.

There were 30 interviews but only two topics that came out of that, designation and incentivisation. How did the farmers understanding of sustainability differ from the expectations of the top-down approach? There are hints of this in the text, but it could be made more explicit or gathered together more. 

The interviewees are identified by letters Ox, A and B. Ox I presume is Ox Mountain, but what does A and B refer to? 

The discussion looks more like the results or results and discussion, the concluding remarks the discussion and the conclusion only one paragraph. 

There is little indication of how the farmers actually monitor their land, how do they agree on the metrics and indicators used? Again there are hints but no details. There is also no indication of what happens when a farmer cannot meet the expectations or fails in achieving the results expected? Particularly important when this happens due to no fault of the farmer's management and very important to consider in a climate change scenario when previous management regimes may not be enough. 

Specific details:
Page 2: "While this continued unchecked for a number of years, in recent times and particularly in the last decade, we have seen efforts to realign where agricultural is going, prompted by a growing concern for habitats..."

Should this be agriculture?

Page 2: "Undoubtedly, agri-environmental policy has been fundamental in creating numerous positive environmental responses to agriculture including growing knowledge of nutrient"

in missing or a comma.

Page 7: A repeat but different interview numbers

Paragraph 1: "if it was farmed with some of the practices that I know it would respond better and I’m under no doubt about that’ (Ox5)"

Paragraph 2“if it was farmed with some of the practices that I know it would respond better and I’m under no doubt about that” (Ox7)."

Reviewer 2 Report

I found this to be a very interesting and relevant paper. I think the author has done an excellent job of teasing out the farmer statements to create a compelling narrative. The writing style is sometimes a little less formal (and stiff) than is usual in academic literature, but I found that added to the strength of the paper rather than distracted from it. I do have a few small suggestions that you might want to consider.

Abstract: There's a typo in the 3rd line of the abstract: agir-environment.

Throughout the paper, the Oxford comma is sometimes used, sometimes not, and sometimes placed after the 'and'. It should either be always or never.

Introduction, first paragraph. "a past of over-exploitation, degradation and pollution and, in more times, a re-emergence of a new appreciation for protection". The word 'recent' is missing between 'more' and 'times'.

Section 2, paragraph 1, last sentence: Extremes of what exactly? I see how thought can vacillate, but don't see how management has done so. 

Section 2, paragraph 2: The statements "Agriculture and agricultural practices have perhaps the most profound impact on the myriad of habitats and species that are found across our rural areas", and "What is equally valid is that in recent decades the incongruity that is farm intensification and land abandonment, has dramatically altered our biodiversity and broader ecosystems", each need a reference.

In paragraph 3, the statement "A key component, and what has often been described as one of the most impressive achievements in the environmental field, were the emergence of agri-environmental policies" could also use a reference.

Section 3, paragraph 1. I think this should be 'populations of bird species'. I don't see how a distribution can decline.

It should be 'have declined substantially' rather than 'declining substantially'.

Section 3, paragraph 2 sentence 2. There's a split infinitive, in case anybody cares.

Section 3.1, sentence starting with "What is also interesting in this process, and a situation that undermines...": I don't think you need to point out that it is interesting. If it wasn't interesting, you wouldn't have written it.

Section 3.1: The following sentence doesn't need the first half of the sentence before 'landowners'.

Section 3.3, first sentence: The word 'expectation' is unnecessary.

Section 3.3, sentence starting with "EIP-Agri..." needs a comma after 'sectors'. Otherwise, this sentence is very hard to understand.

Section 4: "The Burren is a particular fragile landscape...", 'Particular' should be 'particularly'.

I would call section 5: 'Results and discussion'.

Section 5.1, first sentence: I think you can just say the that the process has often been met with conflict.

Page 7, second paragraph. 'were' instead of 'where'.

Page 7, second paragraph. Last sentence. I think there's a copy paste mistake here. This quote has already been stated in the previous paragraph, but attributed to a different farmer.

Section 5.1, last paragraph, last sentence. i think there are some words missing in this sentence.

Section 5.2. Only the first paragraph of this section is about incentives. The rest is about participation. You might want to reconsider the subheading.

Section 5.2, last paragraph, first sentence: Should be 'fewer' rather than 'less'.

Section 6.2. Should be 'demand' rather than 'demands'.

Reviewer 3 Report

Title: "Designation, Incentivisation and farmer participation – exploring options for sustainable rural landscapes"

  1. The manuscript falls within the “Sustainability” journal’s aims and scope
  2. The topic is important (but generally local). The article in the introduction does not justify the importance of the research for countries other than Ireland.
  3. The abstract is fairly well organized but should also contain straightforward information about the purpose, method, and sources of the research.

I suggest making some improvements based on my questions and comments (above and below).

  1. What theory underlies the current research?
  2. The paper has failed to provide the concepts of “sustainable landscape” and “sustainable rural landscapes”. It should need to consult some additional works and/or propose its own specific concept. What is a common understanding of “sustainable landscape”? I strongly suggest referring to published papers such as, for example:
  • Marc Antrop, 2006. Sustainable landscapes: contradiction, fiction or utopia?, Landscape and Urban Planning, 75 (3–4), 187-197, https://doi.org/10.1016/j.landurbplan.2005.02.014
  • Paul Selman, 2008. What do we mean by sustainable landscape?, Sustainability: Science, Practice and Policy, 4 (2), 23-28, DOI: 1080/15487733.2008.11908019
  • Adrian Southern, Andrew Lovett, Tim O’Riordan, Andrew Watkinson, 2011. Sustainable landscape governance: Lessons from a catchment based study in whole landscape design, Landscape and Urban Planning, 101 (2), 179-189, https://doi.org/10.1016/j.landurbplan.2011.02.010.
  • MaÅ‚gorzata Luc, 2014. Placing the Idea of Sustainable Landscape in Ecophilosophy Problemy Ekorozwoju – Problems of Sustainable Development, 9 (1), 81-88, https://ssrn.com/abstract=2387292
  1. What is the research aim and objective?
  2. Methods: How many individuals (farmers, landowners) were interviewed and when? Are they representative of the main study area (“the Burren and the region around the Ox Mountains”, the Western region of Ireland)?
  3. The manuscript lacks the “Results” section. The results are included in the Discussion section, so the fifth section could eventually be titled "Results and Discussion".
  4. What exactly are lessons from Ireland "in terms of how we might envision sustainable rural landscapes going forward are utilized"? Are they presented in concluding remarks?
  5. How has sustainability science contributed to the current research and how can the current research contribute to sustainability science?
